# A Methodological Approach to Assess the Impact of Energy and Raw Materials Constraints on the Sustainable Deployment of Light-Duty Vehicles by 2050

**Mihai Machedon-Pisu** *[ID] **and Paul Nicolae Borza** [ID]

Department of Electronics and Computers, Transilvania University of Brașov, B-dul Eroilor nr. 29, 500036 Brașov, Romania; borzapn@unitbv.ro
\* Correspondence: mihai_machedon@unitbv.ro

**Abstract:** Light-duty vehicles represent the land transport means with the most prominent impact on environment, society's travel needs, and market dynamics. The evolution of different powertrains is analyzed herein mainly in terms of the raw materials sensitive to exploitation and the energy use in three stages: production, operation, and end of life. In this sense, this study proposes a methodology based on balancing the rapports between supply and demand in order to evaluate every powertrain's market share by 2050. The results of this analysis are compared to the outputs of other models and frameworks that aim to assess the sustainable deployment of transport means. The results show that scenarios that propose a market share of 25% for battery electric vehicles are unlikely to happen by 2050 due to the disruptions of the lithium, cobalt, and nickel supply chains, while the ambitious target of 50% market share for battery electric vehicles is not possible by then. The main findings of this study refer to the role played by battery chemistry and storage capacity in determining the market penetration of various powertrains for light-duty vehicles under the specific constraints of the automotive sector related to energy and materials.

**Keywords:** automotive sector; sustainable deployment; energy; raw materials; storage

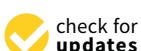



## 1. Introduction

Land transport activities play an essential role within our society. At the same time, these activities generate directly huge amounts of pollutants due to vehicle operation, and indirectly, more pollution due to vehicle production and recycling as well as a significant consumption of energetic resources and raw materials [1]. A raw image of sustainability for the land transport activities is illustrated in Figure 1. The land transport market is dominated by light-duty vehicles (LDVs) and will still be until 2050, both in terms of energy use and passenger travel demand [2]. Powered by the corresponding legislation and specific policies in the automotive sector, various roadmaps, frameworks, models, and life cycle assessment (LCA) studies have been assessed at the European level in order to develop sustainable transport systems [3–8]. Their starting point is represented by EU transport reports and statistics [9,10]. In such analyses, the medium- and long-term impacts of energy supply and demand balancing and regulations play an important role in shaping the appropriate strategies [11].

The design of such strategies should tackle the discrepancies between the use of energy and raw materials and the sustainability goals of transport systems in a timely fashion, whether such an approach is event-based, and thus occasional, or is continuous and consequent, based on time sampling. The traditional model for sustainable transport emphasizes mainly three components: Economy, Society, and Environment, as presented in the framework in [12]. A more simplified model is illustrated in [13].

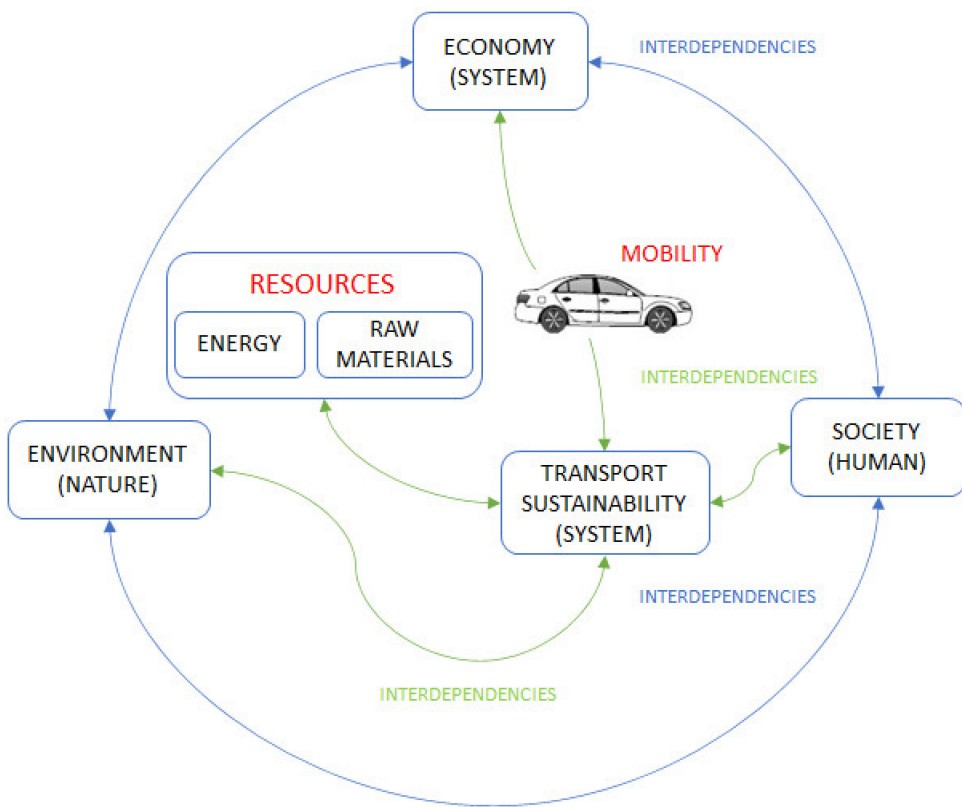

**Figure 1.** Simplified sustainable transport model including energy and raw materials inferences.

The main components of sustainability consist of Energy, Technologies for transport, Resources, the Environment, and the Society/Economy, as seen in Figure 1. The Energy component includes aspects related to the production and efficiency in the generation and consumption of energy in vehicles. The Transport-related technologies category covers aspects related to the use of better fuels and infrastructure as well as associated implications on transport. The Resources component refers to the demand of raw materials and energy, as well as energy storage issues. The Environment component includes the generation of transport related emissions at each stage and associated impacts on health and environment. The Socio-Economic component covers issues related to the market and its implications on transport. The automotive market is in demand of special policies based on consumer's behavior. Behavior always plays a significant role. For example, the acceptance of new vehicles such as EVs is many times followed by financial benefits such as incentives, subsidies, pollution taxes, etc.

The resources are split between energy and other raw materials in order to emphasize the essential role played by them in the overall balance of generated pollutants and, of course, on the equilibrium between the efficiency and the utility of the transport activities. For the production loop, the main aspect refers to the technological aspect, as a result of the scientific progress in this particular field. This loop also emphasizes the constraints of the ecologic consequences resulting from the development of transport activities, in synchronism with the demands generated by society. The modulating role of the strategies and policies adopted by society aims to overdrive the natural evolution of the system.

The operation activities are influenced by regulations and the effects on nature and human society. The effects on health and the depletion of natural resources represent the main constraints. The natural resources have a finite character and must be efficiently used during their normal life span and recycled as much as possible after. It also represents at the same time the desiderata of circular economy. Energy is a universal resource, and it is used in the two stages of related activities: infrastructure and operation processes. Energetic resources refer to waste and generation in usage and have important consequences on

all sustainability components [14]. Among the most important are the pollutants and depletion of Earth's resources. An essential role in this system is played by technologies, as human society's know-how and capability to provide improved and more efficient transport means, in order to sustain the transport activities, such as their operation, by optimizing the related processes. The operationalization of the transport means will provide services that implement the desired mobility. Both in the case of production and operation of transport means, energy is consumed. Even by considering the energy resources as unlimited, the generation and consumption of energy produces pollution that must be reduced.

The main path to make this possible consists in increasing the energy efficiency in production and operation. The raw materials incorporated in the transport infrastructure will be totally or partially recovered and re-introduced within a new industrial cycle (desiderata of circular economy) [15]. The society's effort to satisfy its needs must be feasible and bearable, and this aspect is closely related to the sustainability of related activities. The activities are triggered mainly by the increasing needs of society for transport activities.

Looking back into the car's history, one can notice how their technological development was correlated with the socio-economic, environmental, and health context, and has paved the way for new policies and strategies for exploiting the available energy sources and materials with respect to the current automotive regulations. Figure 2 presents the major milestones and impact on energy resources.

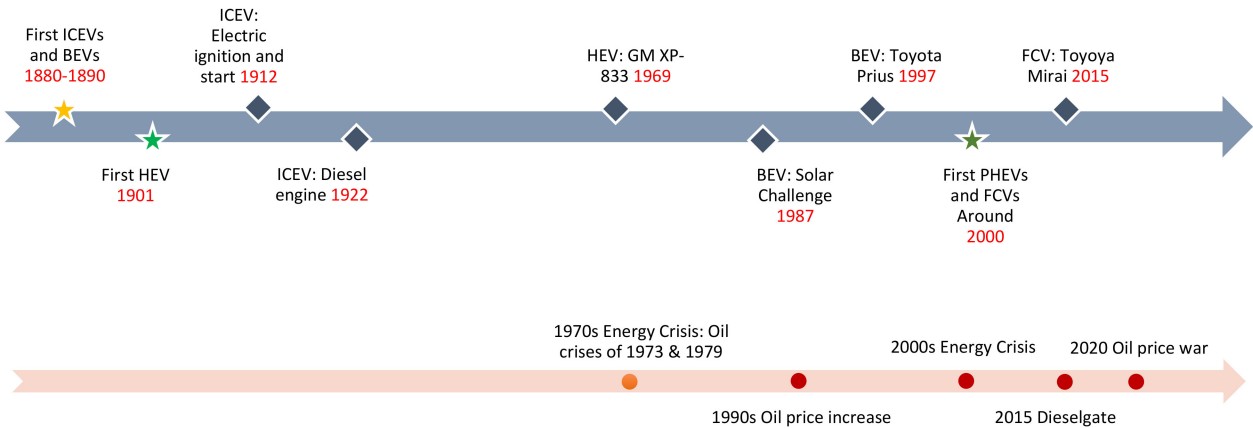

**Figure 2.** Major milestones in the history of cars and impact on energy.

The modern vehicle, first developed by Karl Benz in the 1880s, is based on the internal-combustion engine (ICE). The first battery electric vehicles (BEVs), equipped with a DC power system and energized by lead–acid batteries, competed with ICEVs in the 1890s. Electric vehicles (EVs) were less noisy, did not encounter the starting problems of the ICE, and had no tailpipe emissions. In addition, the low range of the BEVs was not considered a problem at that time. In 1901, Ferdinand Porsche invented the first hybrid electric car, the Lohner-Porsche Mixt. Yet, EV sales were to collapse over the next decade, as the manual crank was eliminated by Charles Kettering's invention of the electric ignition and start. The diesel engine was introduced in 1922 as a more efficient compression–ignition (CI) IC engine compared to the spark–ignition (SI) IC engine fueled by gasoline. Hybrid driving cars made a comeback in 1969 thanks to the GM XP-883 model of General Motors (GM). In 1973, a new hybrid car prototype was proposed due to pollution issues. In 1997, Toyota developed the Prius, which was the first mass production hybrid car.

In the late 1980s, GM decided to develop an all-electric car. One of the reasons was the urban pollution of the American cities, especially Los Angeles. Another reason was the success of the solar-powered Sunraycer electric car in the Solar Challenge race in 1987 [16]. In recent decades, there have been new vehicle proposals as alternatives to the vehicles mentioned above. Plug-in hybrid electric vehicles (PHEVs) were developed at the end

of the 21st century when William H. Patton developed the first hybrid boat propulsion system. The first Fuel Cell vehicle (FCV) was manufactured by GM under the name of Electrovan. Fuel cell vehicles have been commercially available only since 2001 [17]. In addition, new efforts have aimed to reduce the battery dependency of EVs by using hybrid electric storage systems (HESS). In this regard, the electric storage can be split coherently between supercapacitors (SC) and batteries [13,18].

The paper's motivation consists of the need to highlight the degree of resilience of the evolution mechanism of transport technologies conditioned by the conversion from classical to electric vehicles. The main objective is to obtain herein a holistic picture of this transition. On the one hand, the associated growing needs and the ways in which they can be met are analyzed, and on the other hand, the consequences of the evolution relative to material and energy resources are detailed. In addition, aspects regarding the quality of life (related to caused pollution, impacts on health and ecosystems) and society's ability to withstand this evolution in terms of economic efficiency and generated added value are presented. The questions to be answered are related to the capacity of the transport system to develop and function as a system capable of permanently balancing supply and demand. The proposed methodology aims to find out mainly the potential space for sustainable evolution of the technological transition of LDVs, from ICE vehicles to electrical cars, by identifying and analyzing the potential scenarios specific to this evolution.

The content of the paper is as follows. After the introduction in Section 1, which evokes the evolution of the transport domain with the main events, the paper shortly defines the sustainability framework depicted in the next sections. In Section 2, an overview of the technological progress in the automotive sector is portraited. Such an overview is addressed also in the context of the legislation's evolution, which is mainly illustrated by EU decisions that could lead to the Green Deal program. Section 3 includes the proposed methodology that covers the production, operation, and end of life stages, which are typical to a Life Cycle Assessment (LCA) analysis, and compares it other sustainable transport models and frameworks. In Section 4, two extreme scenarios that forecast the LDV market share in terms of use and balancing of the demand and supply in materials and energy, Low Exploitation Rate (LER) and High Exploitation Rate (HER), are analyzed for one snapshot: 2050. The results are discussed in the final section, which highlights the main findings of this study.

## 2. Technological and Legislative Impacts on the Automotive Sector Development

### 2.1. Overview of the Technological Progress and Implications on Resources and Policies

The transport activities are mainly modulated by the evolution of technologies and science in the automotive domain. In the case of transport means evolution, such as cars, this is influenced by the technology that implements an essential component: the powertrain (ICEV, HEV, PHEV, FCV, BEV).

As a result of the evolution of technologies, there is a change in the demand for raw materials and energy. The operation efficiency is highlighted by the structural changes of the transport means, which is generally considered under the dome of "powertrain" technologies, with effects on the primary energy source: fossil fuels, electrochemical sources (FCV and BEV), and renewables.

The main performance metrics of vehicles are closely related to the on-board energy storage capacitance, type, and energy efficiency of the vehicle's energy converter [17]. An important parameter is life span, which is below 10 years [19] for new vehicles such as EVs. On the other hand, as illustrated in Figure 3, the life span of light and heavy vehicles has increased worldwide to 11–13 years [20]. In addition, there is an increase in the weight of the vehicles, and as a consequence, heavy vehicles last 1–2 years more than light vehicles in Europe [21]. However, this implies also the use of more materials and energy resources. At the same time, as shown in Figure 4, the number of cars has increased substantially, as well as vehicle mileage [9,16,22,23].

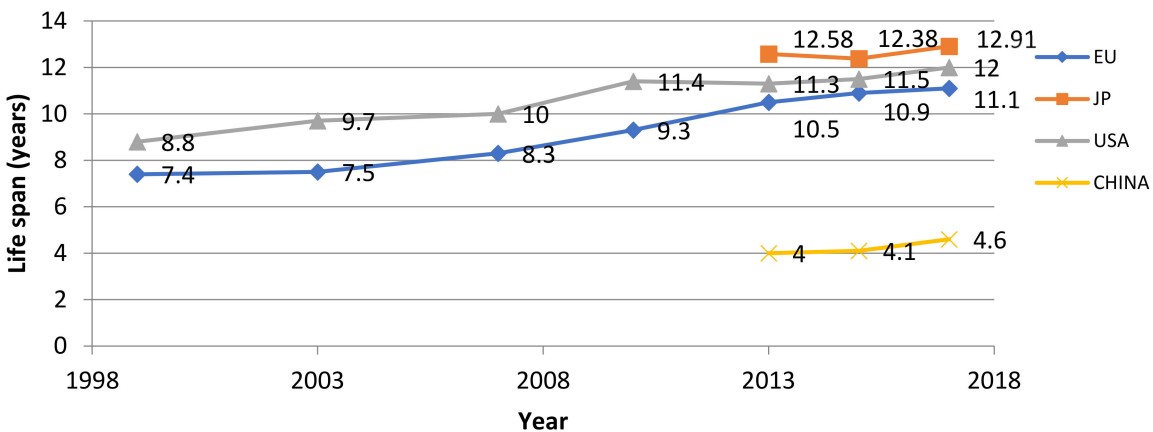

**Figure 3.** Trends in the vehicles' life span in the main world zones.

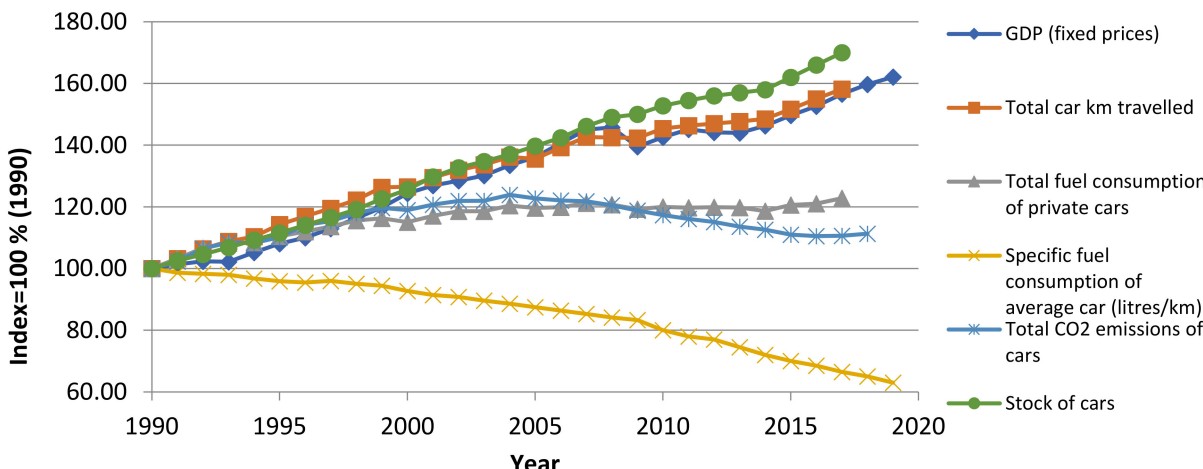

**Figure 4.** Various performance metrics for vehicles during 1990–2019 in Europe.

On one hand, as a consequence of better car technologies, pollution due to greenhouse gases (GHG) was tempered, on the other, unfortunately due to such a GDP-driven evolution of vehicle sales, a substantial reduction of the pollution produced by transport systems will not be possible in the long term. Although the energy efficiency of engines and vehicles has in general improved, the power demand of cars in relation to their allowed consumption has also increased significantly, also as a result of legal constraints. This has led to improved comfort offered by vehicles, as well as to better dynamic characteristics. There is a clear improvement in the operation efficiency of vehicles, but pollution mitigation has a limited effect due the increase in the number of vehicles in circulation. Thus, the pollution curve cannot follow this significant positive evolution in energy efficiency.

The growth in GDP is also associated to the substantial increase in population and thus in the demand of energy and resources. Figure 5 illustrates the evolution of fuel resources (supply and demand of oil), as seen in [24]. Such an image, such as the one presented in [25], includes business-as-usual (BAU) assumptions for the future evolution in the use of oil resources and passenger cars. In addition, as a direct consequence of human activities—for example, the growing number of passenger cars sold per year, as presented in [26]—the change in land–ocean temperature is as dramatic as ever, leading to a highly negative environmental impact. Recently, starting from 2015, the Paris Climate Agreement recommended that the world must follow a path consistent with a 2 °C stabilization scenario in order to get a grip on global warming. As presented in [27], global emissions would not decline in absolute terms relative to 2015 levels let alone meet this scenario by 2030.

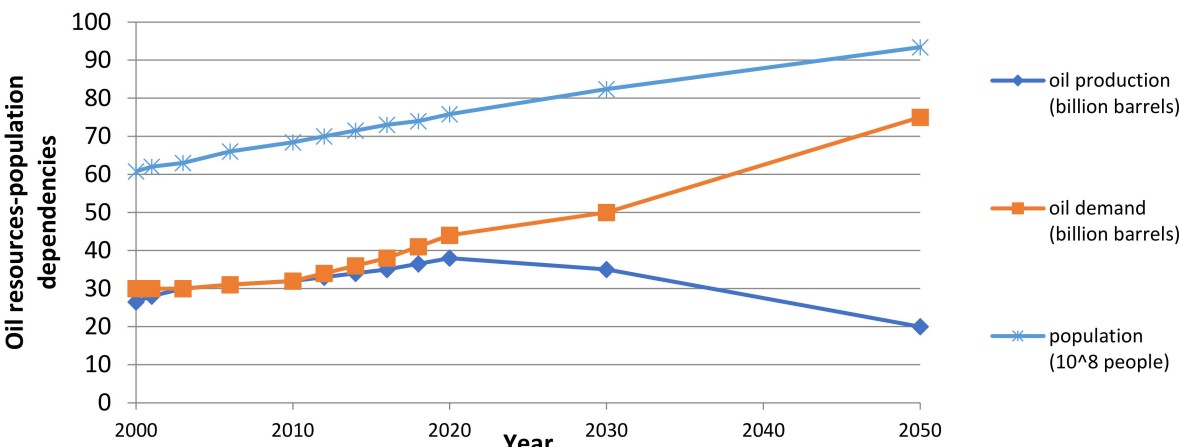

**Figure 5.** Evolution of worldwide oil resources depending on world population dynamics: oil demand and production.

The society's support for such climate goals, such as green energy, is of utter importance, and personal involvement plays an important part, as shown in [28]. The new European Union (EU) Green Deal, which aims to make Europe the first climate-neutral continent by 2050, should reveal the importance of taking the necessary actions to protect the soil by using tools such as Biodiversity Strategy and European Climate Law, as stated in [29]. Another study on the EU Green Deal [30] shows that current policies will achieve only 60% reductions in GHG by 2050 from 1990 levels, which is not sufficient to comply with the Paris Agreement. Therefore, more efforts are necessary in this sense, and so the Green Deal must accelerate the shift toward sustainable transport strategies that could result in quantifiable health benefits.

### 2.2. Overview of the European Standards and Implications on Environment and Health

Due to technological reasons (e.g., lack of infrastructure, reduced energy storage, low autonomy) and economic reasons (e.g., high property costs), BEVs, PHEVs, and FCVs have not yet managed to spread sufficiently on the market. The use of resources, which are related to the energy source and raw materials, and the consequences on the environment, are also of utter importance in this sense, especially in terms of human health. Unfortunately, only a small portion of primary energy sources are renewable [27].

Some exceptions refer to cases where clean energy is a relevant option, such as Norway and California, wherein societal efforts have been broad and better targeted. One can notice that the welfare of citizens plays a significant role, too. Alongside market considerations, one should also consider the industrial pollution that results from EV's and FCV's manufacture and electricity use. Unlike conventional cars that pollute more while being operated than when being produced, EVs pollute significantly also in the industrial stages [1].

More than over 200 years of industrialization took a toll on the environment and population's health, and nowadays, the climate change problem is as stringent as ever. Moreover, air pollution due to industrial activities can cause serious damage to human health and environment [31], especially in terms of premature deaths [32]. In this regard, various legislative measures in all sectors producing air pollutants, such as GHG, have been proposed by the EU in [33]. According to the World Health Organization (WHO) [34], in 2016, 90% of the urban population was exposed to Particulate Matter (PM) concentrations that exceeded the maximum allowed values. In this respect, EU has adopted a comprehensive set of directives that follow most of the WHO guidelines for reducing the levels of PM, NOx, $O_3$, $SO_2$, CO, and benzene, among others [33]. These pollutants are compared in Figure 6a. According to [33], the main activities that cause pollution can be associated with domestic, industrial, electricity generation, and transport sectors, as estimated by the European Commission for Environment. As seen in Figure 6b, one of the main sources that contribute to air pollution is road transport.

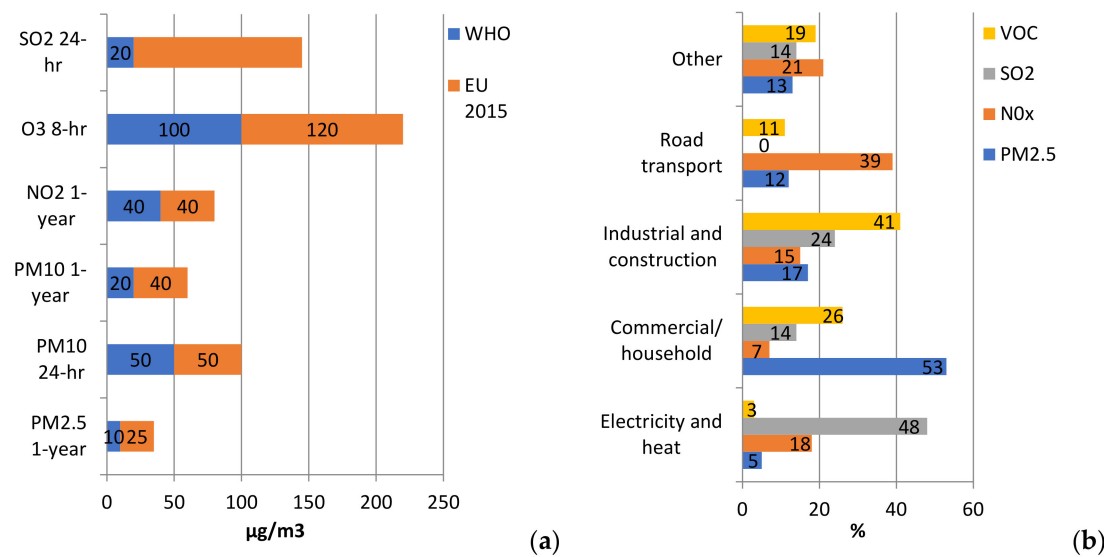

**Figure 6.** The main air pollutants classified according to: (**a**) maximum allowable values as defined by WHO and European Commission on Environment guidelines [33,34]; (**b**) main activity domains impact in Europe, as estimated in [34].

In terms of legislative practices at the European level, one should emphasize the steps undertaken in the direction of the management and the improvement of air quality, according to the EU directives [33]. The European emission standards for the automotive sector, such as passenger cars (Category M) and light commercial vehicles (Category N1 class III and N2), were developed circa 1993 and the latest Euro 6 standard targets the year 2021. The allowable limit values (g/km) for vehicles on diesel and gasoline refer to PM, NOx, and CO [35]. The reported values for PM2.5, PM10, NOx, CO, $SO_2$, $O_3$, Pb, benzene, and benzo-a-pyrene (BaP) at the European level start from 1999 until 2017, according to [36].

It is worthwhile to notice the evolution of stricter emission standards and air quality values at the European level, especially after 2011. Yet, it must be mentioned also that the values based on WHO guidelines have always been stricter, as seen in Figure 7. In fact, the latest European air quality reports show a significant discrepancy between countries that respect the latest European limit values in terms of PM2.5, PM10, and BaP but fail to meet the WHO guidelines. One must think about the recent automotive scandal from 2015 [37] and understand the need for imposing stricter limits on the air pollution resulted from transport activities, as depicted in Table 1.

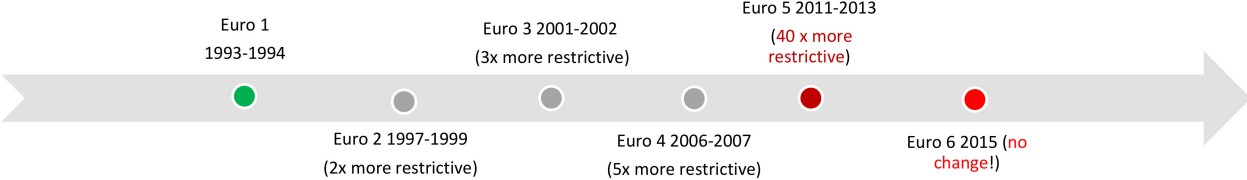

**Figure 7.** Evolution of Euro automotive regulations.

**Table 1.** NOx limit and measurement values (g/km) for two Volkswagen car models.

| Model | United States | | Euro 5–6 | | |
| --- | --- | --- | --- | --- | --- |
| | Limit | West Virginia University Measurement | Limit Euro 5 | Limit Euro 6 | Measurement 2011 |
| Jetta | 0.043 | 0.61–1.5 | 0.18 | 0.08 | 0.62 ± 0.19 |
| Passat | 0.043 | 0.34–0.67 | - | - | 0.62 ± 0.19 |

According to [36], it is also possible to illustrate the evolution of premature deaths attributable to PM2.5, NO$_2$, and O$_3$ within EU27/28 between 2011 and 2016 for a population varying from 502,960,000 to 510,180,000 inhabitants. In terms of the population's health, this can be seen as a stagnation, which can be attributed to the lack of stricter limits implemented at the European level after 2011. There are only minor improvements: 14.2% for PM2.5, 10.5% for NO$_2$, and 14.6% for O$_3$, when comparing the values reported for 2016 to the values for the 2011–2015 period. As an important remark, considering only the pollution associated with the car's operation as the sole or most relevant metric when opting for 'cleaner' electric vehicles can be very misleading. Similar statements were advertised by car manufacturers regarding diesel ICEVs, and the consequences in the past are known [37]. Other aspects such as human toxicity and other potentials [38] are just as important, as soil and water pollution also play an important role in defining both environmental and health risks, besides the more visible air pollution aspects.

## 3. Proposed Methodology for Assessing the Overall Impact of Energy and Raw Materials Constraints on the Sustainable Deployment of LDVs Based on Market Share Outputs

The vehicle market share outputs of the sustainable transport models and frameworks presented in [12,22,39–42] can be synthesized in Figure 8 in terms of two extreme scenarios. The ICE-based scenarios are considered conservative, as is the case of the BAU scenario in [39], while the EV-based scenarios are considered progressive, as is the case of the Base scenario in [41].

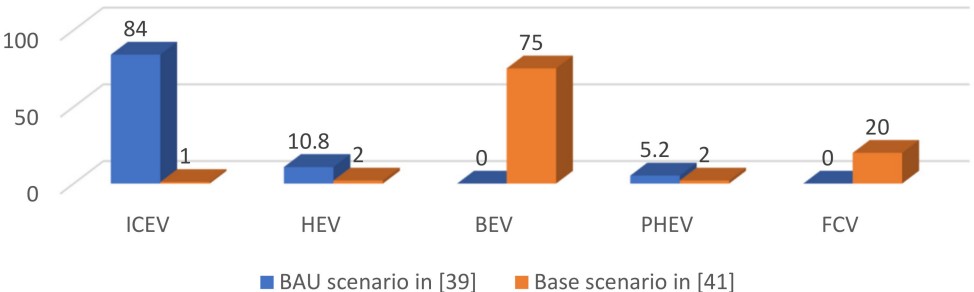

**Figure 8.** 2050 Vehicle market share according to a conservative and a progressive scenario/prediction.

The rationale behind our methodology comes from the fact that the various models and frameworks proposed for sustainable transport in [3–7,10,12,22,39–42] mostly omit addressing the interactions that occur at each stage, from production to end of life, according to a Life Cycle Assessment (LCA)-based analysis [43], especially in terms of resources associated to the use of raw materials and energy for producing, operating, and recycling LDVs, as presented in Figure 9.

The mentioned models are analyzed in Table 2 in terms of supply and demand balancing factors for raw materials and energy. As can be seen, no model can satisfy more than half of the requirements depicted in Table 2 for the four supply/demand balancing factors.

The proposed methodology based on the LCA procedure, similar to the one in [43], is presented in Figure 9 in simplified form.

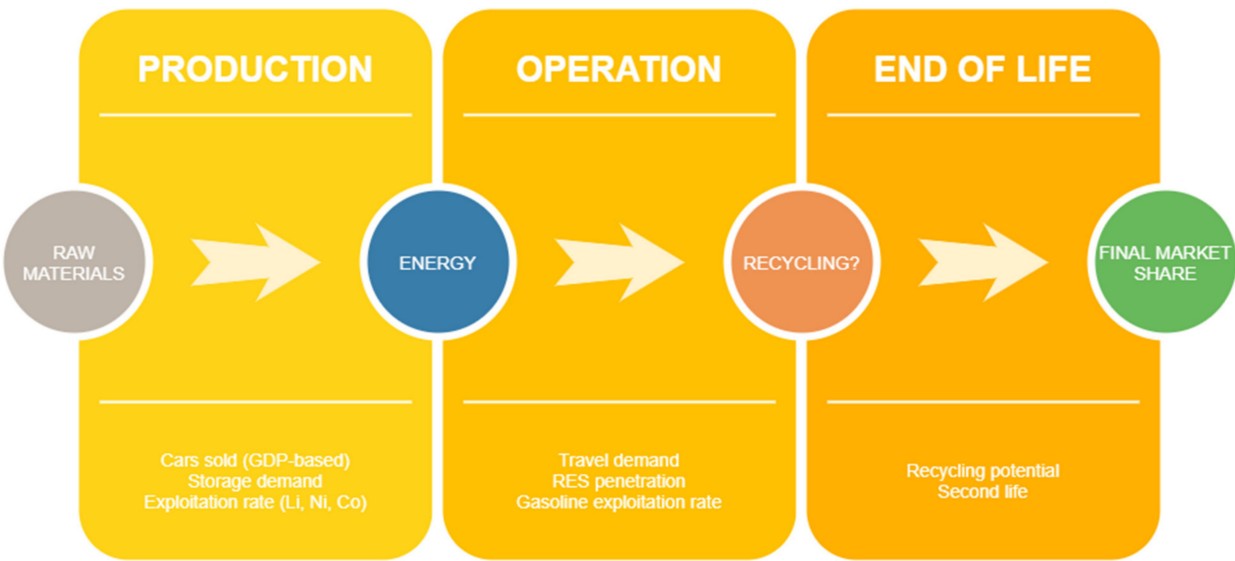

**Figure 9.** Simplified LCA-based methodology based on energy and raw materials inferences.

**Table 2.** Analysis of studied models and frameworks in terms of raw materials and energy supply/demand imbalances.

| Supply/Demand Balancing Factors | Models in [41,42] | Models in [39,40] for Framework in [11] | POLES Model [3] for Enerdata in [10,22] | GEM-E3 [4] and PRIMES-TREMOVE models [5–7] | This Study |
|---|---|---|---|---|---|
| Exploitation of raw materials: Li, Co, Ni, Pt | None [41,42] | None [39,40] | None [3] | None [4], [5–7] | Yes |
| Fuel production in gasoline, electricity production | None [41,42] | None [39,40] | Yes [3] | Yes [4], [5–7] | Yes |
| Raw materials demand and storage demand due to battery chemistry | Yes, but no storage demand analysis [41], none [42] | None [39,40] | None [3] | None [4], [5–7] | Yes |
| Fuel consumption in gasoline, electricity consumption | Yes, but no RES analysis [41,42] | None [39], only gasoline consumption [40] | Yes [3] | Yes [4], [5–7] | Yes |

## 4. Analysis of the Proposed Methodology in Terms of LDV Market Share Outputs

As depicted in the methodology described in Figure 9, both the predicted number of cars sold and the storage demand in terms of sensitive raw materials can decide the EVs market share. On one hand, population dynamics [44] and financial status, as seen in Figure 4, determine the evolution of car sales [45,46]. In addition, the increase in Evs acquisition [47] can model the overall demand depicted in Figure 10.

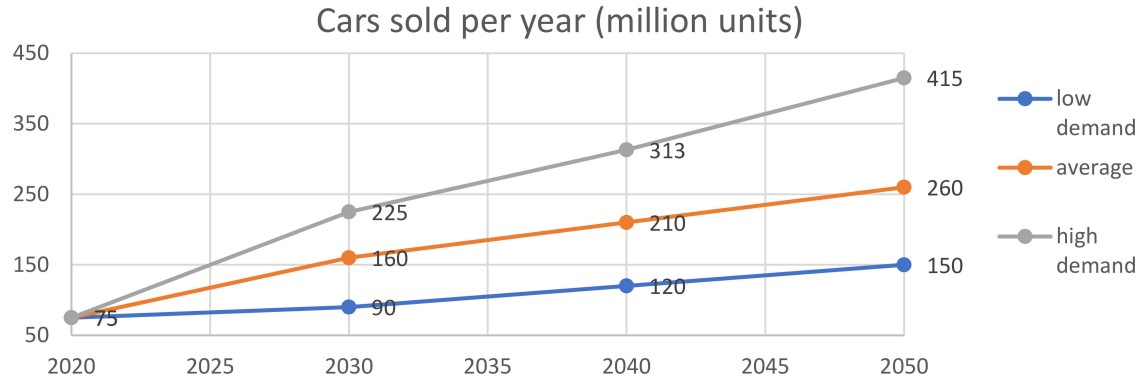

**Figure 10.** Forecast for cars sold per year, from 2020 to 2050.

On the other hand, such growing demand must be sustained by supplying new resources associated with raw materials and energy. In terms of raw materials, the most sensible to exploitation are lithium (Li), cobalt (Co), nickel (Ni), and platinum (Pt) [48,49]. Other sensitive raw materials such as iron (Fe), aluminum (Al), manganese (Mn), copper (Cu), and phosphorous (P) [41,49] must be mentioned too, but due to lesser impact on the supply chain, they were not analyzed herein. In addition, the geopolitical context must be taken into account [14,48]. United States Geological Studies (USGS) provides an annual report that forecasts the available reserves [50]. Such predictions for Li [49–53], Co [48,50], Ni [50,54], and Pt [50] reserves vary from a pessimistic (low supply) to an optimistic forecast (high supply) [50]. The exploitation of predicted Li reserves follows two main evolutions in Figure 11: a low exploitation rate (LER) scenario, which is based on [55] and approximated by a 4th-order polynomial equation, and a high exploitation rate (HER) scenario, which is based on [23] for Li reserves predictions, as justified in [49,56]. These evolutions do not include the recycling potential of raw materials, which will be considered in stage three (end of life).

**Figure 11.** Exploitation of current lithium reserves for the two exploitation rate scenarios.

According to the LER and HER scenarios presented in Figure 11, if no new Li reserves are found, then these reserves could last until the 2100–2136 period for LER and until 2100–2120 for HER, starting from the pessimistic (low supply) to the optimistic (high supply) predictions. According to the LER [50,57] and HER [23,50] scenarios presented in Figure 12, if no new Co reserves are found, then these reserves could last until the 2077–2168 period for LER and until 2080–2120 for HER, starting from the pessimist (low supply) to the optimistic (high supply) predictions.

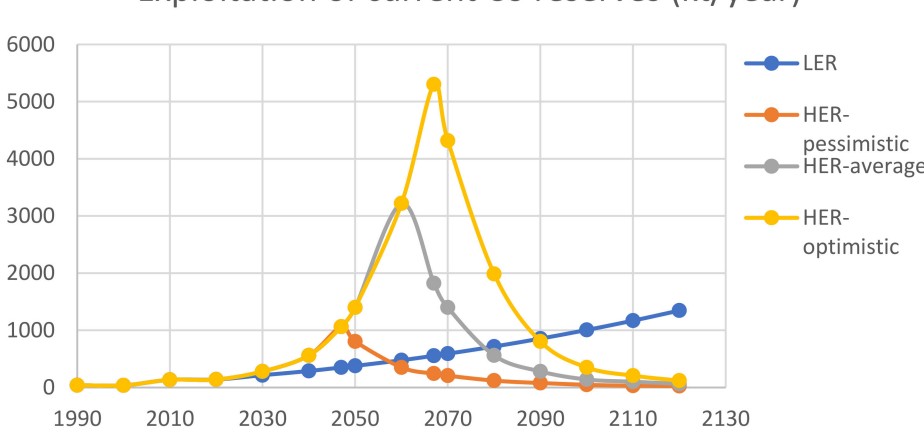

**Figure 12.** Exploitation of current cobalt reserves for the two exploitation rate scenarios.

According to the LER [23,58,59], and HER [23,50] scenarios presented in Figure 13, if no new Ni reserves are found, then these reserves could last until the 2045–2074 period for LER and until 2031–2043 for HER, starting from the pessimist (low supply) to the optimistic (high supply) predictions.

## Exploitation of current Ni reserves (kt/year)

**Figure 13.** Exploitation of current nickel reserves for the two exploitation rate scenarios.

According to the LER [23,60,61] and HER [23,50] scenarios presented in Figure 14, if no new Pt reserves are found, then these reserves could last until the 2129–2149 period for LER and until 2110–2130 for HER, starting from the pessimist (low supply) to the optimistic (high supply) predictions.

Not all Li, Co, Ni, and Pt reserves can be solely used by LDVs, as other applications require also considerable amounts, as seen in Table 3. Actually, how much of these reserves will be used by LDVs is still very uncertain. Various studies try to give some insights on how much of the Li [62–66], Co [63,67,68], Ni [69,70], and Pt [71,72] reserves can be used by cars. These predictions are synthesized in Table 3.

## Exploitation of current Pt reserves (t/year)

**Figure 14.** Exploitation of current platinum reserves for the two exploitation rate scenarios.

**Table 3.** Materials and energy demand of LDVs in terms of use in 2050.

| Raw Materials | High Use | | Low Use | |
|---|---|---|---|---|
| | Other Applications | LDVs | Other Applications | LDVs |
| Lithium | 30% | 70% | 60% | 40% |
| Cobalt | 20% | 80% | 50% | 50% |
| Nickel | 60% | 40% | 90% | 10% |
| Platinum | 40% | 60% | 70% | 30% |
| Gasoline | 60% | 40% | 80% | 20% |

In addition to the supply chain issues caused by the manner in which the sensitive raw materials are exploited, other issues associated with the storage demand of cars affected by battery chemistry and capacity constraints are just as relevant, especially in the case of EVs. NMC/NCM, NCA, LFP, and Pt-based batteries are already technologically available, while the same cannot be said about Li-based batteries such as Li-S and Li-Air. Starting from the battery chemistry, it is possible to determine the need in sensitive raw materials. This depends also on the car storage capacity in case of EVs and is highlighted in Table 4. These values can be approximated to the ones found in [49,73–75]. Most BEVs have a battery size that varies from 50 to 100 kWh [62,76]. Due to the higher energy densities of NMC and NCA batteries compared to the much lower density of LFP, as seen in [62], LFP will be solely used for lower battery size BEVs such as 50 kWh. In addition, new battery chemistries based on Li-S and Li-Air [77] can satisfy the higher energy demand of BEVs with 100 kWh battery size. Starting from the methodology presented in Figure 9, the authors analyze how the market share can respond to the raw materials constraints at stage one (production) and stage three (end of life) and energy constraints at stage two (operation).

**Table 4.** Storage demand of LDVs.

| Battery Type | Lithium (kg/kWh) | Cobalt (kg/kWh) | Nickel (kg/kWh) | Platinum (kg/vehicle) |
|---|---|---|---|---|
| NMC/NCM | 0.133 | 0.32 | 0.435 | - |
| NCA | 0.242 | 0.142 | 0.79 | - |
| Li-S | 0.412 | - | - | - |
| Li-Air | 0.136 | - | - | - |
| LFP | 0.168 | - | 0.01 | - |
| Platinum-based | - | - | - | 0.046 |

The demand in raw materials is correlated with the storage constraints, and by consulting Table 3, NMC/NCM and Li-Air battery chemistries are analyzed herein for use in 100 kWh BEVs, while LFP is analyzed for 50 kWh BEVs. Stage one (production) does not take into consideration the recycling potential associated with stage three (end of life, with 100% recycling). Yet, the no-recycling tag seen in Figures 15–17 does not mean that the EV's batteries cannot be used at the end of their service life. Second life applications are possible as long as the lower battery capacity (70–80% from the initial capacity) can satisfy the application's requirements. The market share that results from the supply/demand calculations is presented in Figures 15 and 16 for 100 kWh BEVs and in Figure 17 for 50 kWh BEVs and Pt-based FCVs.

Stage two (operation) deals with energy constraints in terms of gasoline reserves for ICEVs, HEVs, and PHEVs running on gasoline (high travel autonomy) and of available renewable energy (RES) for charging BEVs and PHEVs running on stored energy in batteries (low to medium travel autonomy). The starting point represents the predicted travel demand in passenger km [2,78], as depicted in Figure 18.

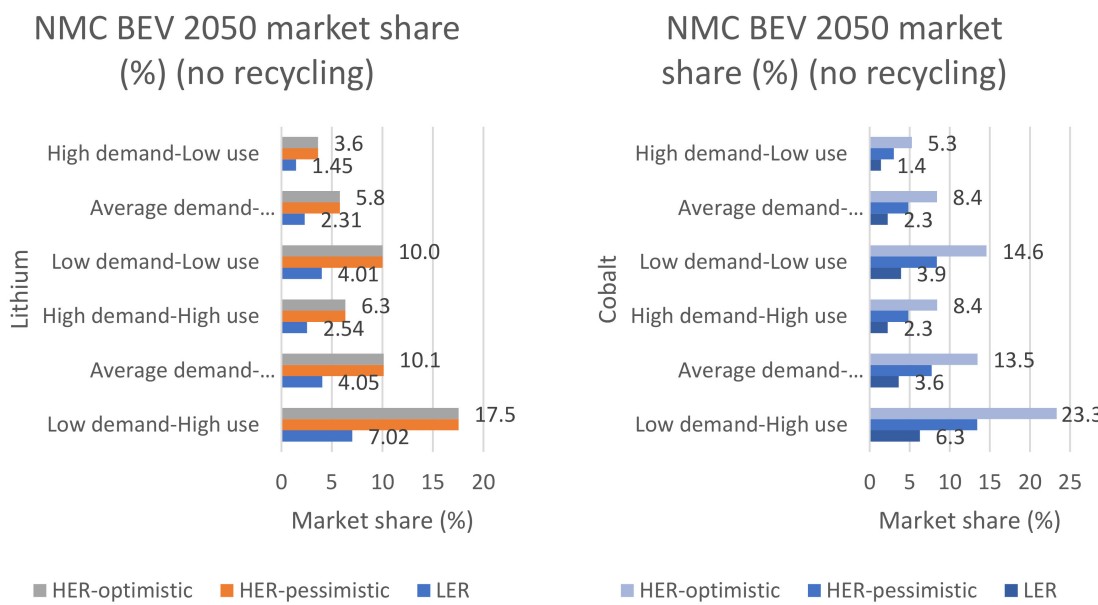

**Figure 15.** 2050 market share for 100 kWh NMC BEVs based on Li and Co reserves, in terms of exploitation of supply, demand, and use (after stage one, no recycling).

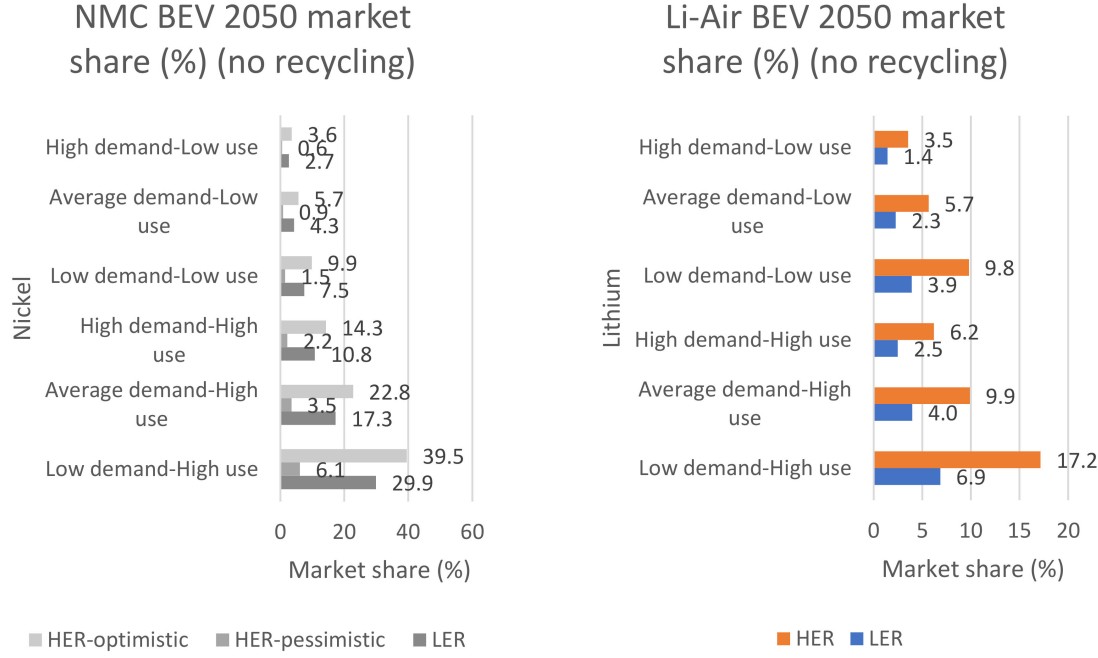

**Figure 16.** 2050 market share for 100 kWh NMC BEVs based on Ni reserves and for 100 kWh Li-Air BEVs based on Li reserves, in terms of exploitation of supply, demand, and use (after stage one, no recycling).

By applying a similar evolution to the one in Figures 11–14, the gasoline consumption per year in liters per gasoline equivalent (LGE) follows two evolution curves by 2060, in Figure 19, that aim to reduce the current high consumption of ICEVs and HEVs, which is 5535 billion LGE [79] to around 30% (1600 billion LGE) for a LER prediction, based on [25], which means that current reserves can last up to around 70 years until they run out, and for a HER prediction, based on [24], which means that current reserves can last to around 40 years. Other fuels such as bioethanol, diesel, CNG, and hydrogen are analyzed in terms of demand in [41].

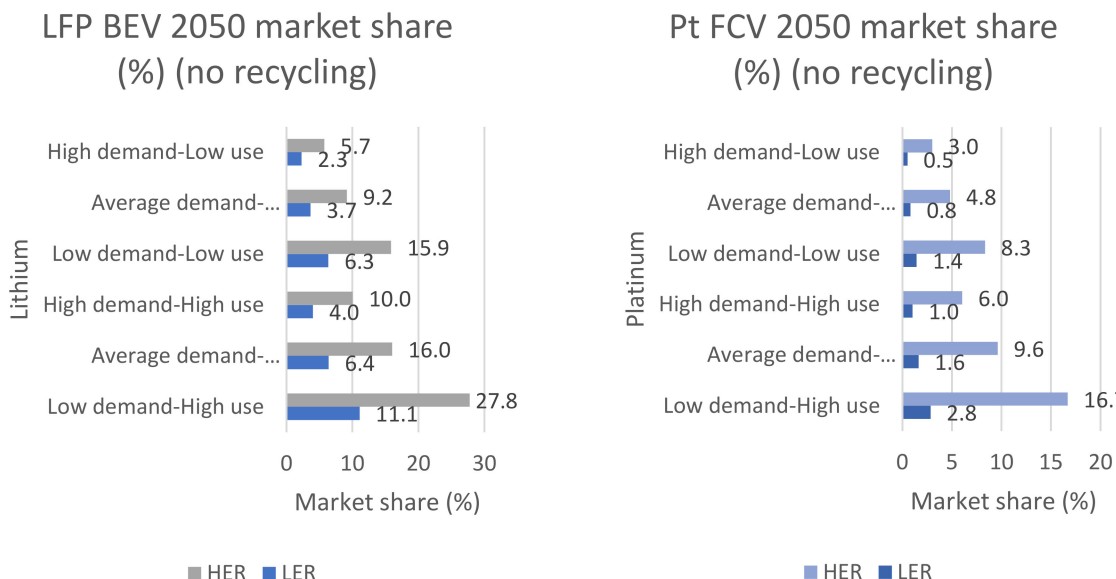

**Figure 17.** 2050 market share for 50 kWh LFP BEVs based on Li reserves and for platinum-based FCVs based on Pt reserves, in terms of exploitation of supply, demand, and use (after stage one, no recycling).

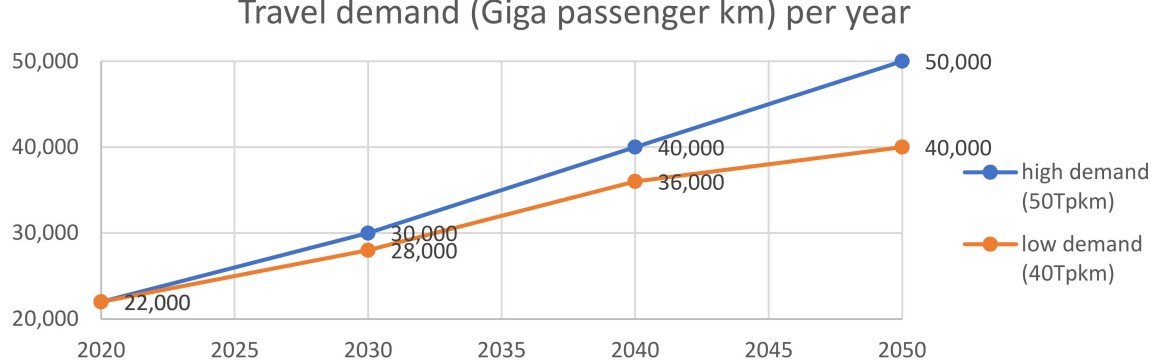

**Figure 18.** Forecast for travel demand from 2020 to 2050.

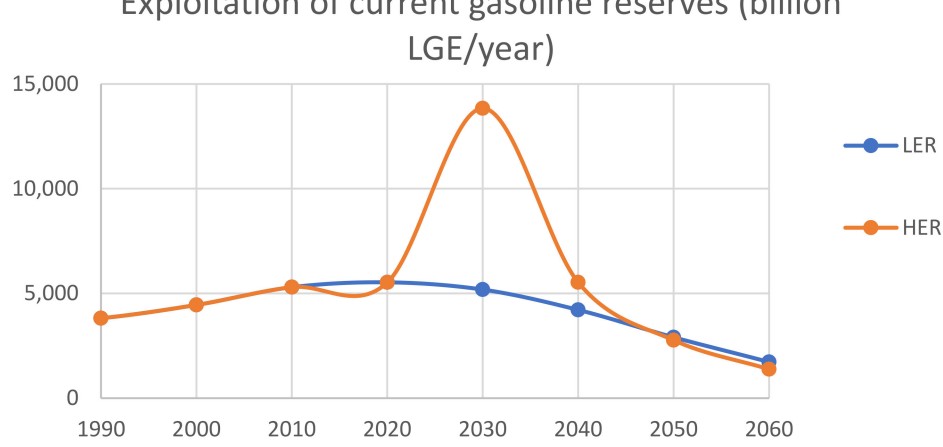

**Figure 19.** Forecast for gasoline consumption for the two exploitation rate scenarios.

Since not all the gasoline can be used to fuel LDVs, the predictions for gasoline use in cars can vary from 20% (low use) to 40% (high use) according to various studies [80–82].

How much clean energy is available for charging EVs is depicted in Figure 20, according to [83–85] for optimistic (high supply) and pessimistic (low supply) predictions, as well as the total energy available predicted by 2050.

### Available energy (TWh) per year

Legend:
- RES s+w_pessimistic
- RES s+w_optimistic
- Total_pessimistic
- Total_optimistic

**Figure 20.** Forecast for available energy for charging EVs from 2020 to 2050.

The resulting market share for gasoline dependent LDVs is depicted in Figure 21.

Assuming only clean energy (RES) is used for charging EVs, the market share is also determined in Figure 21 for EVs based on supply/demand calculations for the two use scenarios, according to [86,87].

Stage three (end of life) takes into consideration the recycling potential of the raw materials discussed in stage one, according to [77]. The predicted decrease in materials is presented in Figure 22. The adjusted market share is presented in Figures 23 and 24 for Li and Pt reserves. In addition to the raw materials mentioned in the production stage, such as Li, Co, Ni, Mn, Al, and Cu [41,49], which are all candidates for the recycling stage, also silicon (Si) and graphite must be considered for this final stage, as depicted in [71].

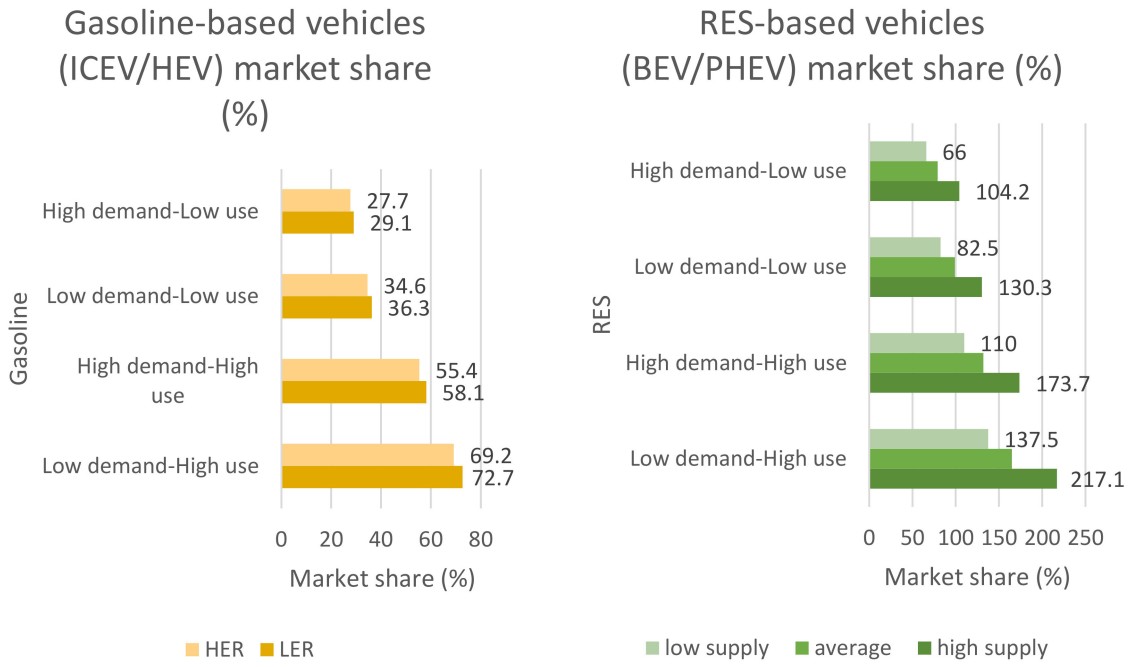

**Figure 21.** 2050 market share for ICEVs/HEVs/PHEVs-on fuel based on gasoline reserves and for BEVs and PHEVs-on electricity based on RES for charging, in terms of supply/demand and use (stage two).

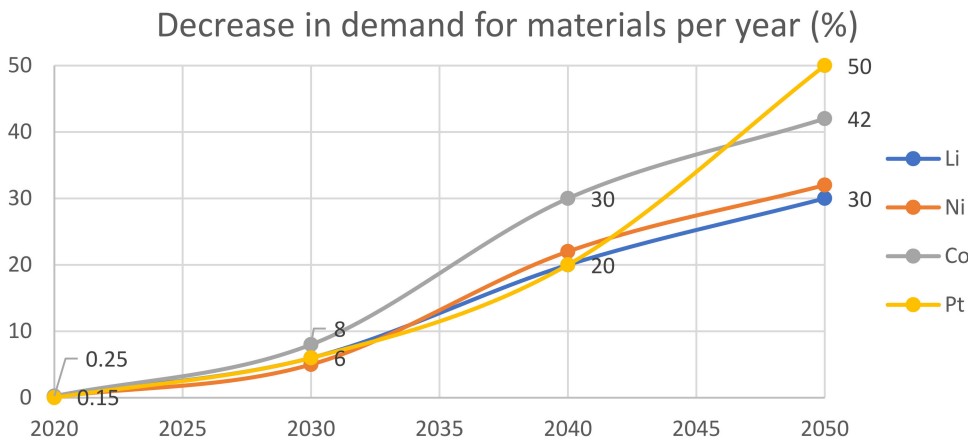

**Figure 22.** Forecast for recycling potential, from 2020 to 2050.

After the three stages, by considering an average supply/demand ratio, the market share for all LDV types is depicted in Figure 25 for NMC-based BEVs with a capacity of 100 kWh and in Figure 26 for Li-Air-based BEVs with a capacity of 100 kWh and for LFP-based BEVs with a capacity of 50 kWh, for the two use scenarios.

As seen in Figure 25, the high use case follows the more conservative scenario, which is comparable to the ones in Figure 8, while the low use case follows the more progressive scenario in Figure 8, leaving the door open for non-platinum FCVs and lower battery capacity EVs. As seen in Figure 21, charging EVs is not an issue in terms of RES demand, since it is unlikely that their market share will exceed 66%.

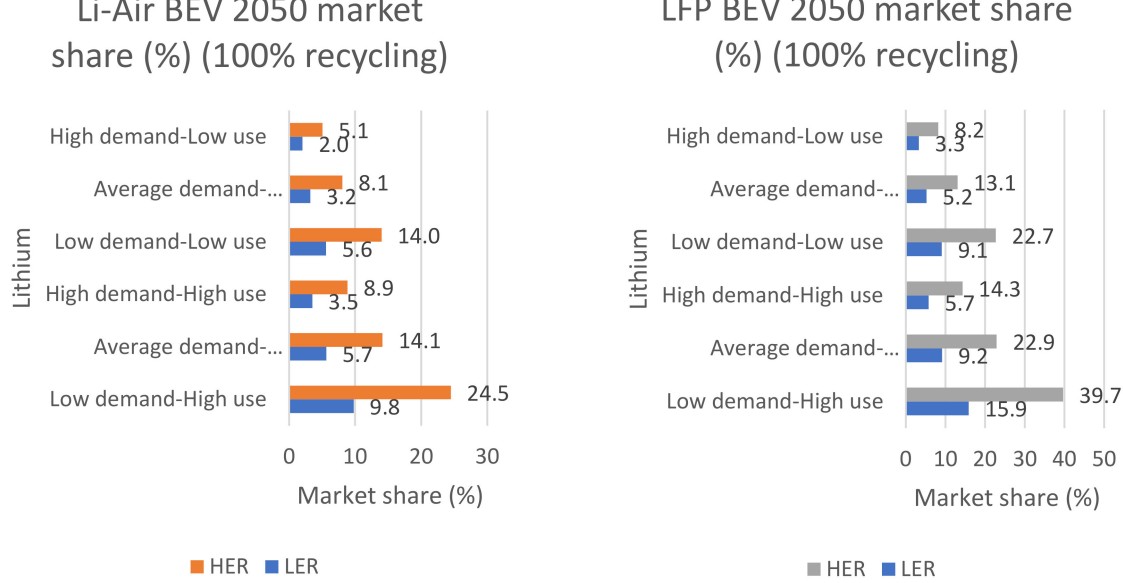

**Figure 23.** 2050 market share for 100 kWh Li-Air BEVs and 50 kWh LFP BEVs based on Li reserves, in terms of exploitation of supply, demand, and use (after stage three, 100% recycling).

As seen in Figure 26, the high use case follows the more conservative scenario which is comparable to the ones in Figure 8, while the low-use case follows the more progressive scenario in Figure 8, leaving the door open for non-platinum FCVs and lower battery capacity EVs. For meeting the Stated Policies (STEP) requirements (25% market share for BEVs), it is imperative to decrease the storage capacity. As seen also in Figure 26, only LFP-based BEVs can get near to such requirements. In the high use case, the market share follows a more likely conservative scenario, while in the low use case, the progressive scenario is more noticeable.

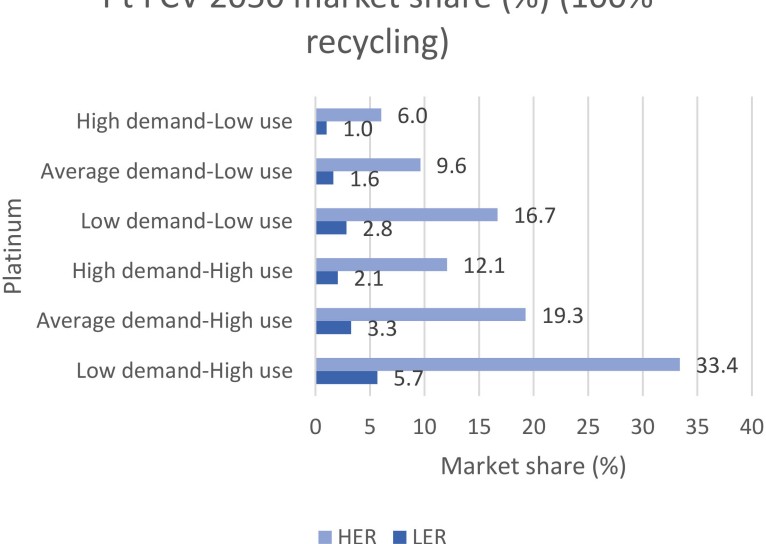

**Figure 24.** 2050 market share for Platinum-based FCV based on Pt reserves, in terms of exploitation of supply, demand, and use (after stage three, 100% recycling).

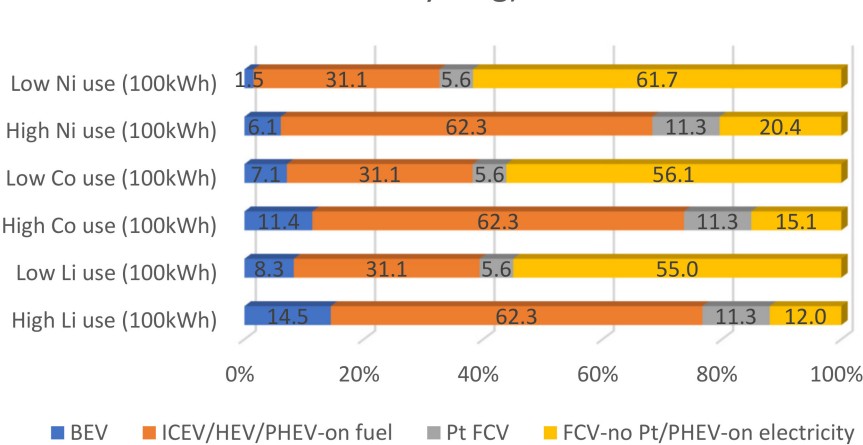

**Figure 25.** LDV 2050 market share based on average supply–demand ratio only for NMC 100 kWh BEVs (after stage three, 100% recycling).

The LDV market share outputs are synthesized in Tables 5–7 for the three main scenarios: worst-case scenarios for BEVs (BEV WCS), average supply/demand scenarios, and best-case scenarios (BEV BCS). WCS refers to a minimum value of the supply–demand ratio, thus for low supply and high demand, and it usually refers to a LER evolution. BCS refers to a maximum value of the supply–demand ratio, thus for high supply and low demand, and it usually refers to a HER evolution, especially for the optimistic case.

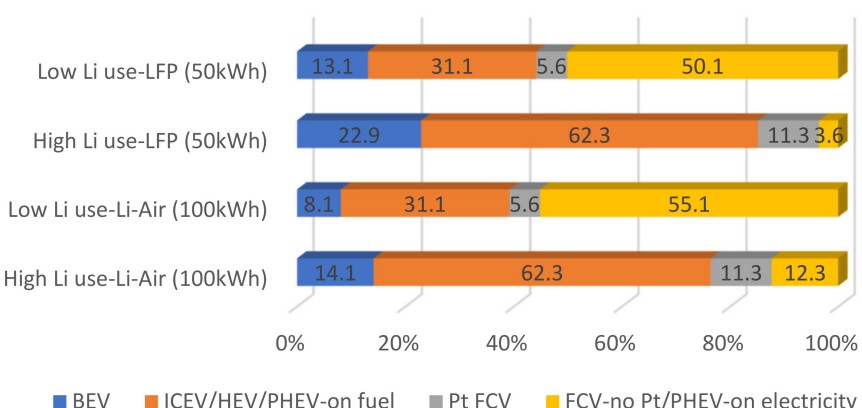

**Figure 26.** LDV 2050 market share based on average supply–demand ratio for Li-Air 100 kWh BEVs and LFP 50 kWh BEVs (after stage three, 100% recycling).

**Table 5.** Market share for NMC, Li-Air, and LFP worst-case scenarios for BEVs (BEV WCS), 100% recycling.

| Scenario | BEV Market Share (%) | ICEV/HEV/PHEV-Gasoline Market Share (%) | Pt FCV Market Share (%) | FCV-No Pt, PHEV-No Gasoline Market Share (%) |
|---|---|---|---|---|
| NMC Li-high use | 3.62 | 55.35 | 2.84 | 38.18 |
| NMC Co-high use | 3.34 | | | 38.47 |
| NMC Ni-high use | 3.80 | | | 38.00 |
| Li-Air-high use | 3.54 | | | 38.26 |
| LFP-high use | 5.74 | | | 36.07 |
| NMC Li-low use | 2.07 | 27.68 | 1.42 | 68.83 |
| NMC Co-low use | 2.09 | | | 68.82 |
| NMC Ni-low use | 0.95 | | | 69.95 |
| Li-Air-low use | 2.02 | | | 68.88 |
| LFP-low use | 3.28 | | | 67.62 |

**Table 6.** Market share for NMC, Li-Air, and LFP average supply/demand scenarios, 100% recycling.

| Scenario | BEV Market Share (%) | ICEV/HEV/PHEV-Gasoline Market Share (%) | Pt FCV Market Share (%) | FCV-No Pt, PHEV-No Gasoline Market Share (%) |
|---|---|---|---|---|
| NMC Li-high use | 14.46 | 62.27 | 11.27 | 12 |
| NMC Co-high use | 11.06 | | | 15.4 |
| NMC Ni-high use | 5.03 | | | 21.43 |
| Li-Air-high use | 14.14 | | | 12.31 |
| LFP-high use | 22.89 | | | 3.57 |
| NMC Li-low use | 8.26 | 31.13 | 5.64 | 54.96 |
| NMC Co-low use | 6.91 | | | 56.32 |
| NMC Ni-low use | 1.25 | | | 61.97 |
| Li-Air-low use | 8.08 | | | 55.15 |
| LFP-low use | 13.08 | | | 50.15 |

**Table 7.** Market share for NMC, Li-Air, and LFP best-case scenarios for BEVs (BEV BCS), 100% recycling.

| Scenario | BEV Market Share (%) | ICEV/HEV/PHEV-Gasoline Market Share (%) | Pt FCV Market Share (%) | FCV-No Pt, PHEV-No Gasoline Market Share (%) |
|---|---|---|---|---|
| NMC Li-high use | 24.09 * | 59.86 * | 16.05 * | 0 |
| NMC Co-high use | 30.29 * | 54.97 * | 14.74 * | |
| NMC Ni-high use | 46.32 * | 42.33 * | 11.35 * | |
| Li-Air-high use | 23.69 * | 60.18 * | 16.14 * | |
| LFP-high use | 33.45 * | 52.48 * | 14.07 * | |
| NMC Li-low use | 14.32 | 31.13 | 8.35 | 54.96 |
| NMC Co-low use | 21.45 | | | 56.32 |
| NMC Ni-low use | 17.03 | | | 61.97 |
| Li-Air-low use | 14.01 | | | 55.15 |
| LFP-low use | 22.68 | | | 50.15 |

* Normalized, because total market share exceeds 100%.

## 5. Conclusions and Future Challenges

Overall, this paper provides a holistic image of the transport domain, especially LDVs, in the context of the depletion of raw materials, the growing energy prices, and pollution with major consequences on the sustainable development of society. The rationale of this study consists of addressing the question of whether the society is capable of assuring a smooth transition from the ICE-endowed vehicles to the fully electric transport, which is said to represent a more clean and efficient solution for solving the problem of mobility of people and goods. The implications on environment and implicitly on health at each stage are reiterated in [1,3–11,27–41,88].

As shown in the previous section, STEP scenarios are only possible if the best-case scenarios for BEVs (BEV BCS) are feasible, which leads to the assumption that the supply–demand ratio is kept at maximum. The implications on the Li, Co, and Ni supply chains are major, meaning that the depletion of materials could not be covered if no new reserves are discovered. The worst-case scenarios for BEVs (BEV WCS) are the most sustainable from the supply chain's point of view. However, by keeping the exploitation of supply at minimum, the low market share of BEVs would leave the door largely open for FCVs and PHEVs. At the moment, these are far away from being deployed due to infrastructure and market issues. Therefore, the best compromise is found in the average supply/demand scenarios. Regarding SD scenarios, even if the reserves of sensitive raw materials could double by 2050, there is no guarantee that their exploitation rate would follow a similar evolution. One of the main reasons refers to the geopolitical context associated with the mining of those resources.

The results of this study show that battery chemistry and storage capacity play a primary role in determining the LDV's powertrain market share as well as the usage of materials and energy in electric energy storage systems (EESS).

The current study does not target all research efforts in the field of EESS nor does it analyze hybrid systems (batteries and supercapacitors or batteries, supercapacitors, and fuel cells) that will be able to prove themselves as reliable solutions in the near future, with improved reliability in the field of transport. In addition, the role of the correlation between the development of RES and EESS is not detailed herein. This will play an extremely important role in the future of this field. The need for a comprehensive analysis in this sense is justified in [1–12,14–20,26,27,42,43,86–89].

The article once again reiterates the need to form a forward-looking image of transport systems and highlights the possible limits that the associated activities generate in correlation with the technological developments in the field of transport and the depletion of raw materials and energy reserves. The consequences for the evolution of ecosystems in general are revealed. The proposed methodology ensures a framing of this evolution with clear implications in obtaining a certain degree of sustainability for the investigated process.

**Author Contributions:** Conceptualization, M.M.-P. and P.N.B.; methodology, M.M.-P. and P.N.B.; software, M.M.-P. and P.N.B.; validation, M.M.-P. and P.N.B.; formal analysis, M.M.-P. and P.N.B.; investigation, M.M.-P. and P.N.B.; resources, M.M.-P. and P.N.B.; data curation, M.M.-P. and P.N.B.; writing—original draft preparation, M.M.-P. and P.N.B.; writing—review and editing, M.M.-P. and P.N.B.; visualization, M.M.-P. and P.N.B.; supervision, M.M.-P. and P.N.B.; project administration, M.M.-P. and P.N.B.; funding acquisition, M.M.-P. and P.N.B. All authors have read and agreed to the published version of the manuscript.

**Funding:** This research received no external funding.

**Institutional Review Board Statement:** Not applicable.

**Informed Consent Statement:** Not applicable.

**Data Availability Statement:** The data presented in this study are available on request from the corresponding author. The data are not publicly available due to the large data sets on which the data presented in the article were based upon.

**Acknowledgments:** The authors would like to kindly thank Louis Francois Pau and Roxana Matefi.

**Conflicts of Interest:** The authors declare no conflict of interest.

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
