# Peer review of "A Methodological Approach to Assess the Impact of Energy and Raw Materials Constraints on the Sustainable Deployment of Light-Duty Vehicles by 2050"

_sustainability, doi:10.3390/su132111826_

Round 1

Reviewer 1 Report

The paper is very well written. The study made in this paper is highly important considering the electric vehicles for future. The detailed information regarding the market share of the Battery Electric Vehicle gives us an idea of the disruptions of Lithium, Nickel and Cobalt. This study could effectively be used to plan on this for employing other resources as well.

On the whole, good job has been done by the authors.

Author Response

Point 1: The detailed information regarding the market share of the Battery Electric Vehicle gives us an idea of the disruptions of Lithium, Nickel and Cobalt. This study could effectively be used to plan on this for employing other resources as well.

Response 1: Thank you kindly for your review. Your suggestions were very useful, and in consequence, we have introduced the items required related to other resources in terms of raw materials and fuels. Please find in the text the changes marked with red (there are three paragraphs).

Reviewer 2 Report

The topic of the manuscript is very interesting, especially for the in-depth analysis provided on the sustainability of batteries and their constituent materials. The topic is well presented, the proposed methodology is rigorous and correctly applied. In my opinion, this manuscript could be published as long as some minor changes are made.

Figure 1 must be improved as it appears blurry.

Figures 15, 16 and 17 could be merged and presented in a more attractive way, the same applies to Figures 18, 19, 20 and Figure 30, Figure 31 and Figure 32.

In the state of the art or in the discussion of the results, at least one paper, dealing with a similar topic, should be compared with what is proposed:

  • Spreafico, C., & Russo, D. (2020). Exploiting the Scientific Literature for Performing Life Cycle Assessment about Transportation. Sustainability, 12(18), 7548.

Among future developments, in my opinion the authors should also specify if and in what way, the data provided could be useful to classify vehicles according to a development trend, along the lines of what has been done for other technologies and following other indicators.

For example, refer to: “Spreafico, C., Russo, D., & Spreafico, M. (2021). Investigating the evolution of pyrolysis technologies through bibliometric analysis of patents and papers. Journal of Analytical and Applied Pyrolysis, 105021.”

Author Response

Thank you kindly for your review. Your suggestions were very useful, and in consequence, we have introduced the items required.

Point 1: Figure 1 must be improved as it appears blurry.

Response 1:  Please find in the text the modified figure.

Point 2: Figures 15, 16 and 17 could be merged and presented in a more attractive way, the same applies to Figures 18, 19, 20 and Figure 30, Figure 31 and Figure 32.

Response 2:  Please find in the text the modified figures, which were initially Figs. 15-32. Now they were compacted into Figs.15-26.

Point 3: In the state of the art or in the discussion of the results, at least one paper, dealing with a similar topic, should be compared with what is proposed.

  • Spreafico, C., & Russo, D. (2020). Exploiting the Scientific Literature for Performing Life Cycle Assessment about Transportation. Sustainability, 12(18), 7548.

Response 3:  Your suggestion was appreciated and we have included it in the discussion of the results, regarding a similar topic in terms of environment and health impact. Please find it marked with red in the final chapter (the first paragraph). The citation was introduced as [88] in References.

Point 4: Among future developments, in my opinion the authors should also specify if and in what way, the data provided could be useful to classify vehicles according to a development trend, along the lines of what has been done for other technologies and following other indicators.

  • Spreafico, C., Russo, D., & Spreafico, M. (2021). Investigating the evolution of pyrolysis technologies through bibliometric analysis of patents and papers. Journal of Analytical and Applied Pyrolysis, 105021

Response 4:  Regarding future developments, your remarks were very useful for us in order to provide a comprehensive perspective on the development trends in this field, similar to the field indicated in the paper suggested. Please find the citation marked with red in the final chapter, which is [89] in References.
